# Impact of Phase Defects on the Aerial Image in High NA Extreme Ultraviolet Lithography

**DOI:** 10.3390/mi16111210

**Published:** 2025-10-24

**Authors:** Kun He, Zhinan Zeng

**Affiliations:** 1School of Electronic Information and Electrical Engineering, Shanghai Jiao Tong University, Shanghai 200240, China; hekunyc@163.com; 2Zhangjiang Laboratory, Shanghai 201210, China

**Keywords:** EUV mask, phase defect, aerial images, high NA, shadow effect

## Abstract

With the development of extreme ultraviolet (EUV) lithography technology to higher numerical aperture (NA), it provides higher resolution imaging quality, which may be more sensitive to the phase defect in EUV mask. Therefore, it is necessary to comprehensively understand the effect of phase defect on the imaging quality depending on the NA. We simulated aerial images of patterned EUV masks for the EUV lithography exposure tool of NA = 0.55 and NA = 0.33 using the rigorous coupled-wave analysis (RCWA) method. The results shows that higher NA enhances the contrast of aerial images, which, in turn, provides greater tolerance for phase defect. This indicates that high NA can mitigate the negative impact of phase defect on imaging quality to some extent. Furthermore, it is found that both the defect signal and the intensity loss ratio of the aerial image first increase and then decrease as the width of the phase defect increases, due to the height/width ratio of the phase defect. Meanwhile, the defect width corresponding to the maximum phase defect signal tends to become smaller as the NA becomes larger. It is also worth noting that when NA = 0.33, variations in the position of the phase defect led to fluctuations in the CD error due to the shadow effect of the absorber, while it diminishes at NA = 0.55. This is because a higher NA of 0.55 provides a stronger background field, which suppresses the shadow effect of the absorber more effectively than it does at NA = 0.33.

## 1. Introduction

The EUV mask is one of the most important components of EUV lithography, and the high-quality EUV mask can ensure the precision and yield of lithography. As the critical dimensions (CD) of chip decreases, EUV lithography technology has become a promising method for 10 nm technology nodes and below [1,2,3,4]. One of the key challenges of EUV lithography is the fabrication of a defect-free mask [5,6,7]. Defects in the EUV mask include amplitude defects and phase defects. The phase defect is mainly from the bumps or pits located under the multilayer or the particles deeply into the multilayer, which can cause the deformation of the multilayer, and ultimately affect the imaging quality. So, phase defect detection poses significant challenges [8,9,10].

Deformed multilayers no longer exhibit ideal mirror reflection characteristics, and there is a significant amount of scattered light in various directions in the reflected light, ultimately reducing the quality of lithographic imaging [11,12,13,14]. It is very important to study the impact of phase defects on aerial images for the development of detection tools and subsequent repair [15,16,17,18,19]. Y. Hao et al. [20] studied how the size and position of phase defects influence the aerial images and the critical dimension (CD) error of line-space (L/S) patterns with the half-pitch (hp) of 22 nm and 33 nm, respectively. They found that the position of the phase defect with the width of 50 nm has a negligible effect for the CD error. But the influence of the position of smaller defects on the CD error needs further investigation. T. Terasawa et al. [21] investigated the effect of the size and the position of the phase defect in the L/S pattern on detection sensitivity. They found that smaller-sized defects cause an increase in the width of the printed pattern, while larger defects cause bridging. However, for the hp 32 nm L/S pattern, the experimental results of the bridging are more significant than simulated results, which needs to be comprehensively studied and analyzed. C. Li et al. [22] investigated the effect of phase defect size on aerial images of different type of absorbers. They found that the aerial image of the L/S pattern was obviously affected by the phase defects with the height from 0.1 nm to 1 nm and the width from 1 nm to 10 nm. In fact, the phase defect will have interference effects with different absorbers. Therefore, it is necessary to further study the influence of the position of the phase defect under different absorbers on aerial images. Y. Kim et al. [23] studied the effect of the size and position of phase defects on aerial images of a square contact hole with the width of 128 nm. It was found that the position accuracy was greatly dependent on the defect position. However, when the width of the phase defect was 92.3 nm, there was a large difference in alignment accuracy between the experiment (almost zero) and the simulation (40 nm) due to the resolution limitation of the contact hole, which relied on the NA of the lithography system. P. Evanschitzky et al. [24] extended a fully rigorous simulation to solve the electromagnetic field and studied the effect of patterned EUV masks with phase defects on the imaging quality. It was found that at NA = 0.25, the hp 32 nm L/S pattern started to bridge at the defect width of 5 nm. At NA = 0.33, the bridge started for a defect top height of 6 nm. It was shown that NA = 0.33 was helpful for reducing the influence of defects on aerial images. V. Schot et al. [25] studied the effect of line width roughness (LWR) on aerial images at high NA, and found that a larger contrast was beneficial to reduce the effect of photon scattering noise on the variation in printed patterns. However, phase defects are essentially different from LWR, and it is necessary to further verify whether the high NA can also alleviate the effect of the phase defect on aerial images. R. Semaana et al. [26] studied the effect of defects on aerial images and found that the lithography system could tolerate larger defects at NA = 0.55. However, the defects under investigation were assumed to be cylindrical structures situated on the mask surface. This assumption may not accurately reflect the true morphology and distribution of phase defects. The impact of Gaussian-type defects buried inside the mask on the aerial image and printed pattern at NA = 0.55 is worth further investigation.

In this paper, a theoretical model of the EUV mask with phase defects and the defect evaluation method are described in Section 2. The result of the impact of phase defect on aerial images and CD error of EUV masks is in Section 3; we first studied the aerial images of the EUV mask without and with phase defects under the conditions of NA = 0.55 and NA = 0.33, respectively. Then, we analyzed the influence of the size of the phase defect on the intensity loss of aerial image and CD error of the printed L/S pattern. Finally, we investigated the impact of the position of the phase defect on the intensity loss of aerial image and CD error. Conclusions are summarized in Section 4. This study is expected to facilitate the development of EUV lithography at higher NA, offering valuable insights for improving accuracy in advanced EUV lithography.

## 2. Theoretical Model and Simulation

### 2.1. Model of the EUV Mask with Phase Defects

The EUV mask is primarily composed of the substrate, the molybdenum/silicon (Mo/Si) multilayer and the absorber. The materials of the substrate and the absorber are SiO_2_ and Ta, respectively. The multilayer is a crucial component for achieving the high reflectivity of the EUV mask, typically consisting of 40 bilayers of Mo/Si, with each Mo/Si bilayer having a thickness of 6.95 nm (Mo/Si = 2.78 nm/4.17 nm). In this simulation, the detailed parameter settings for the EUV mask presented in Table 1.

Defects present on the substrate, as well as those introduced during the multilayer deposition process, can result in multilayer deformation. Figure 1a,b illustrate the pit phase defect and the bump phase defect, respectively. These phase defects interfere with the optical transmission of EUV and degrade the quality of aerial images and the accuracy of printed patterns.

The shape parameters and the location of the phase defect can be expressed as follows:(1)$$z_{top}=H_{top}exp\left(G\frac{{\left(x-x_{pos}\right)}^{2}}{W_{top}^{2}}\right)exp\left(G\frac{{\left(y-y_{pos}\right)}^{2}}{W_{top}^{2}}\right)$$(2)$$z_{bot}=H_{bot}exp\left(G\frac{{\left(x-x_{pos}\right)}^{2}}{W_{bot}^{2}}\right)exp\left(G\frac{{\left(y-y_{pos}\right)}^{2}}{W_{bot}^{2}}\right)$$
where *H*_top_ and *W*_top_ are the height and the full width at half maximum (FWHM) of the phase defect at the top of the multilayer, respectively. *H*_bot_ and *W*_bot_ are the height and the FWHM of the phase defect at the bottom of the multilayer, respectively. *G* is the Gaussian profile factor (*G* = 1 in the simulation). (*x*_pos_, *y*_pos_) is the position coordinates of the phase defect. *z*_top_ and *z*_bot_ are the height distributions of the top and bottom of the multilayer at (*x*, *y*), respectively.

In this paper, we studied a special kind of pit phase defect, which can be expressed as follows:(3)$$H_{top}=H_{bot}$$(4)$$W_{top}=W_{bot}$$

Then, we calculated the diffraction spectrum of the EUV mask by the method of RCWA. In this paper, the mask pattern consists of a periodic array of line features, where directional dependence is negligible. Therefore, an annular source with a center radius of 0.6 and a width of 0.2 is used in the simulation, as well as the configuration that has been widely adopted in many similar studies [28,29,30,31,32]. The detailed parameters of the illumination system are presented in Table 2.

Finally, we accurately simulated the aerial image of the L/S pattern affected by the pit phase defect using the Hopkins model. For NA = 0.33, the reduction factors are 0.25 in both the *x* and *y* directions. For NA = 0.55, the reduction factors in the *x* and *y* directions are 4× and 8×, respectively.

### 2.2. Defect Evaluation

In the simulation, we employed L/S patterns with varying hp. Therefore, the defect signal can be defined as the difference between the aerial images of the L/S pattern with and without phase defects. Generally, when the phase defect is entirely obscured by the absorber, the aerial image intensity is nearly identical to that of the mask without the phase defect, resulting in the disappearance of the defect signal. Figure 2a shows that a pit phase defect can cause significant changes in the aerial image intensity. Since the incident angle is 6°, the distribution of the aerial image intensity is nearly symmetrical in either the *x* or *y* direction. Compared to the peak intensity *I*_0_ of the aerial image without phase defects, maximum intensity *I*_max_ and minimum intensity *I*_min_ of the aerial image appear around the phase defect. The intensity loss Δ*I*_loss_ can be defined as below [12]:(5)$$∆I_{loss}=\frac{I_{max}-I_{0}}{I_{0}}\times 100\left(\%\right)$$

In addition, *I*_0_ can also be defined as the background signal intensity of the aerial image, which refers to the intensity of the non-phase defect area of the aerial image. Likewise, *I*_max_ can also be defined as the defect signal intensity.

Furthermore, as shown in Figure 3b, the presence of a pit phase defect results in pattern deformation, indicating that the printed pattern is affected to a certain degree by the phase defect. The CD error of printed patterns can be defined as deviation from the width of the printed pattern without phase defects by calculating the aerial image intensity and applying a threshold to obtain the real width of the pattern, which can be expressed as follows [12]:(6)$$CD\;error=\left|\frac{S_{PD}-S_{0}}{S_{0}}\right|\times 100\left(\%\right)$$(7)$$S_{0}=hp$$
where *S*_0_ and *S*_PD_ are the width of the printed pattern in the absence of pit phase defects and the width of the printed pattern affected by pit phase defects, respectively.

## 3. Results and Discussion

### 3.1. The Impact of NA on Aerial Images of Defect-Free EUV Masks

The NA of the projection system is very important to the quality of aerial images. We first presented the aerial images at different NA for hp 32 nm and hp 64 nm L/S patterns, as shown in Figure 4. It can be seen from Figure 4e,f that aerial image contrast for NA = 0.55 is larger than that of NA = 0.33. Meanwhile, it can be seen from Figure 4a,b that the contrast of the aerial image of hp 64 nm L/S pattern is larger than that of hp 32 nm L/S pattern.

### 3.2. The Impact of Phase Defect Size on Aerial Images of EUV Masks

Figure 5 shows aerial images of the L/S pattern affected by the phase defect. The phase defect has the height (H) of 1 nm and the width (W) of 20 nm, and is located at the center of the L/S pattern. It can be seen from Figure 5e,f that when NA = 0.55, the difference between the defect signal and the background signal is very weak, because NA = 0.55 has a stronger background signal than that of NA = 0.33, thus having a stronger suppression effect on the defect signal. In addition, it can be seen from Figure 5a,b that the deformation of the printed pattern of hp 32 nm L/S pattern is more obvious, but hp 64 nm L/S pattern is more tolerant to phase defects of the same size.

Figure 6 shows the background signal and the defect signal of different size depending on the NA and the hp of the printed patterns. The signals are plotted as a function of defect position, using curve graphs to clearly illustrate the variation trends. In Figure 6, the red curve represents the defect signal with background, while the blue curve indicates the background. The defect signal can be obtained as the difference between the two curves. It can be seen that the defect signal first increased and then decreased with the defect width, which may be related to the scattering effects caused by the different height/width ratio of the phase defect. For small defects, the large diffraction angle scattered light will be stopped by the pupil, limited by the resolution lithography system. As defect size increases, the diffraction angle decreases, allowing more scattered light to be collected and enhancing the defect signal. For larger defects, some diffracted light then enters the central zone, weakening the phase contrast effect and reducing signal strength [15]. It is worth noting that the width of the phase defect corresponding to the maximum defect signal is smaller for the case of NA = 0.55 compared to NA = 0.33. This is because NA = 0.55 corresponds to a larger pupil. A larger pupil results in a greater diffraction angle for the collection of the scattered light. Consequently, the defect width associated with the peak defect signal will be smaller for NA = 0.55.

Meanwhile, the background signal at NA = 0.55 is larger than that of NA = 0.33. Conversely, the defect signal of NA = 0.55 is smaller, which indicates that the lithography system is more beneficial to suppress the defect signal, thereby reducing the impact of the defect on aerial images.

The relative change in the aerial image is more important for the real lithography process. Figure 7a,b show the ∆*I*_loss_ of aerial images of hp 32 nm L/S pattern and hp 64 nm L/S pattern depending on the size of phase defects. It can be seen from Figure 7 that the ∆*I*_loss_ of hp 64 nm L/S pattern is smaller than that of hp 32 nm L/S pattern, because the larger mask pattern is more inclusive. It can be seen from Figure 7 that the ∆*I*_loss_ at NA = 0.33 is much greater than that of NA = 0.55 for the same size of the phase defect. Meanwhile, the ∆*I*_loss_ first increases and then decreases with the width of the phase defect, which directly correlates with the defect signal variation and shows nearly identical trend behavior, as explained in Figure 6. Moreover, it is worth noting that the height of the phase defect has almost no effect on ∆*I*_loss_. This is because phase defects with a height less than 2 nm exhibit a very small height/width ratio, making the actual reflectivity and phase characteristics primarily dependent on the width of the phase defect [20,34].

More important is the CD error of printed patterns affected by phase defects. As can be seen from Figure 8, the CD error first increases and then decreases with the width of phase defects. The printed pattern of hp 64 nm L/S pattern has a smaller CD error, which indicates a higher tolerance to phase defects than that of hp 32 nm L/S pattern. In addition, it was also found that the CD error at NA = 0.55 is smaller, which indicates that the higher NA lithography system is more tolerant to phase defects of the same size.

### 3.3. The Impact of the Position of Phase Defects on Aerial Images

In this section, we studied the effect of the position of phase defects on the aerial images for the printed patterns at NA = 0.33 and NA = 0.55, respectively. The position of phase defects is shown in Figure 9.

As we known, the position of the phase defect is also important. Figure 10 presents the background signal and the defect signal of different position at NA = 0.33 and NA = 0.55, where the phase defect has the height of 1 nm and the width of 20 nm. It can be seen from Figure 10 that when the phase defect is located at the center of the L/S pattern, both the background signal and defect signal are the largest. Notably, for hp = 32 nm, the background signal is stronger at the mask center due to the overall weaker background (Figure 4). The shadow effect at the mask edges significantly reduces photon collection, leading to weaker local background signals near the edges [20,22]. In contrast, for hp = 64 nm, the much stronger overall background dominates over edge-induced shadow, resulting in a nearly uniform local background signal across the mask. Meanwhile, the background signal is stronger, but the defect signal is smaller at NA = 0.55. This indicates that the ratio of the defect signal to the background signal for higher NA will be smaller.

Figure 11a,b show the relationship between the ∆*I*_loss_ of the aerial image and the position of phase defects of hp 32 nm L/S pattern and hp 64 nm L/S pattern, respectively. As can be seen from Figure 11, when the defect is completely or partially covered by the absorber, the ∆*I*_loss_ is very small. This is because when the defect is covered by the absorber, the illumination light does not pass through the area where the phase defect is located, so the transmission of light is not affected. When the phase defect is located at the center of the L/S pattern, the ∆*I*_loss_ is the largest, and the ∆*I*_loss_ at NA = 0.33 is greater than that of NA = 0.55.

Figure 12a,b show the CD error of printed patterns related to the position of phase defects for hp 32 nm L/S pattern and hp 64 nm L/S pattern, respectively. It is found that CD error of hp 32 nm L/S pattern has a fluctuation near the center of the L/S pattern at NA = 0.33, which may be caused by the shadow effect from the phase defect and the edge of the absorber [20,22]. Meanwhile, the shadow effect is more significant at the edges of the absorber than in the central region. There are likely more scattering and diffraction effects at the edges of the absorber due to the discontinuity of the structure, which may lead to a more substantial shadow effect. However, the validity of this interpretation remains to be confirmed through further research. This shadow effect is influenced by both the aerial image intensity and, more importantly, its specific distribution. Notably, when NA = 0.55, the fluctuation diminishes, and the maximum CD error is reduced compared to that at NA = 0.33. This phenomenon is attributed to the stronger background field provided by a higher NA, as shown in Figure 10. This stronger field effectively mitigates the shadowing effect caused by the absorber, thereby indicating an enhanced tolerance to phase defects of the same size under a higher NA. For hp 64 nm L/S pattern, the phase defect moves from both sides to the center, and CD error increases gradually, reaching a maximum at the center of the L/S pattern. Furthermore, the maximum CD error was smaller compared to that in the hp 32 nm L/S pattern. This was also attributed to the stronger background field provided by the wider L/S pattern, as shown in Figure 10.

## 4. Conclusions

High NA EUV lithography can provide an aerial image with higher contrast, which can reduce the defect effect to some extent [34,35,36,37,38,39]. But for phase defects, the optical effects associated with EUV masks become more complex. Therefore, it is very important to clearly understand the effect of phase defects on aerial images and printed patterns during high NA EUV lithography.

In this work, firstly, we studied the relationship between the defect signal and defect width at NA = 0.33 and NA = 0.55, respectively. It was found that the maximum defect signal at NA = 0.55 is smaller, indicating a higher tolerance to the size of the phase defect. Furthermore, it is noteworthy that the width of defect corresponding to the maximum defect signal is smaller at NA = 0.55. This is because NA = 0.55 can effectively collect scattered light with larger diffraction angles, which was caused by smaller phase defects. Then, we studied the intensity loss of the aerial images and the CD error of the EUV mask affected by phase defects. It was found that both the ∆*I*_loss_ and CD error first increased and then decreased with the width of the phase defect, which is also related to the scattering effect caused by the height/width of the phase defect.

Finally, we investigated the effect of phase defect position on the defect signal, intensity of the aerial images, and CD error. It is found that when the phase defect moves from the two sides of the L/S pattern to the center, both the defect signal and ∆*I*_loss_ increases gradually, which indicates that phase defects at the center of the L/S pattern has the greatest impact on aerial images. What’s more, when NA = 0.33 and hp 32 nm L/S pattern, the CD error exhibits a double-peak variation as the defect moves across a line pattern. This variation may be caused by the optical shadow effect of the absorber. However, when NA = 0.55, the CD error gradually increases as the defect moves toward the center of a line pattern, indicating that a higher NA can effectively suppress the shadow effect.

## Figures and Tables

**Figure 1 micromachines-16-01210-f001:**
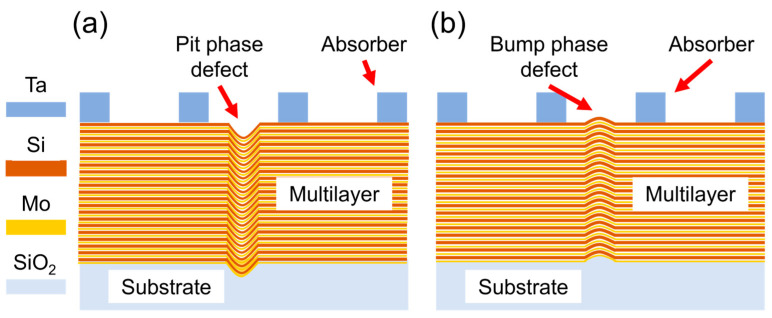
Schematic of the EUV mask with phase defects. (**a**) The pit phase defect, and (**b**) the bump phase defect.

**Figure 2 micromachines-16-01210-f002:**
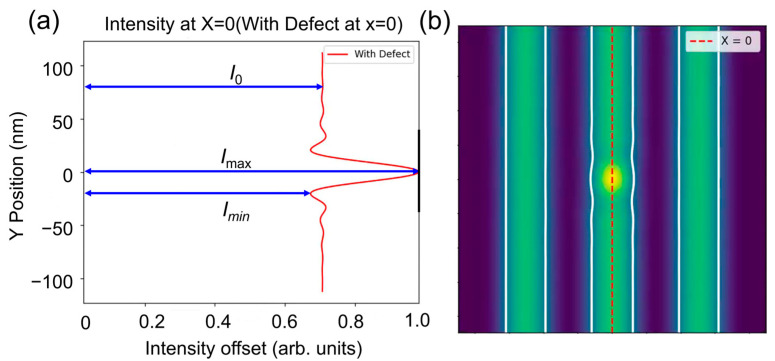
(**a**) The aerial image intensity at x = 0 and (**b**) the aerial image in the x-y plane.

**Figure 3 micromachines-16-01210-f003:**
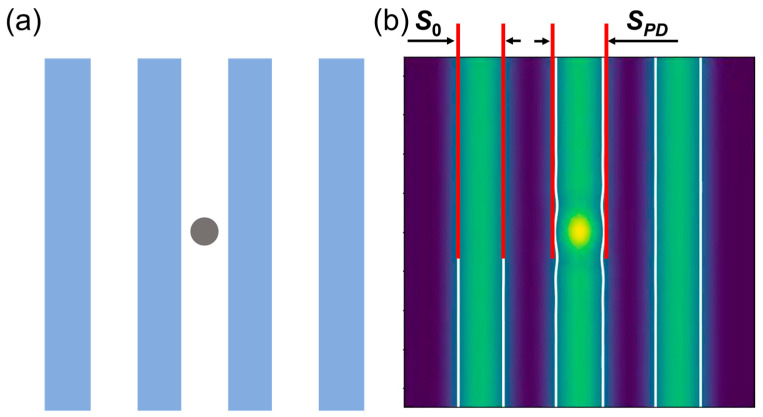
(**a**) The pit phase defect is located at the center of the L/S pattern, (**b**) the variation in the width of the printed pattern due to the pit phase defect.

**Figure 4 micromachines-16-01210-f004:**
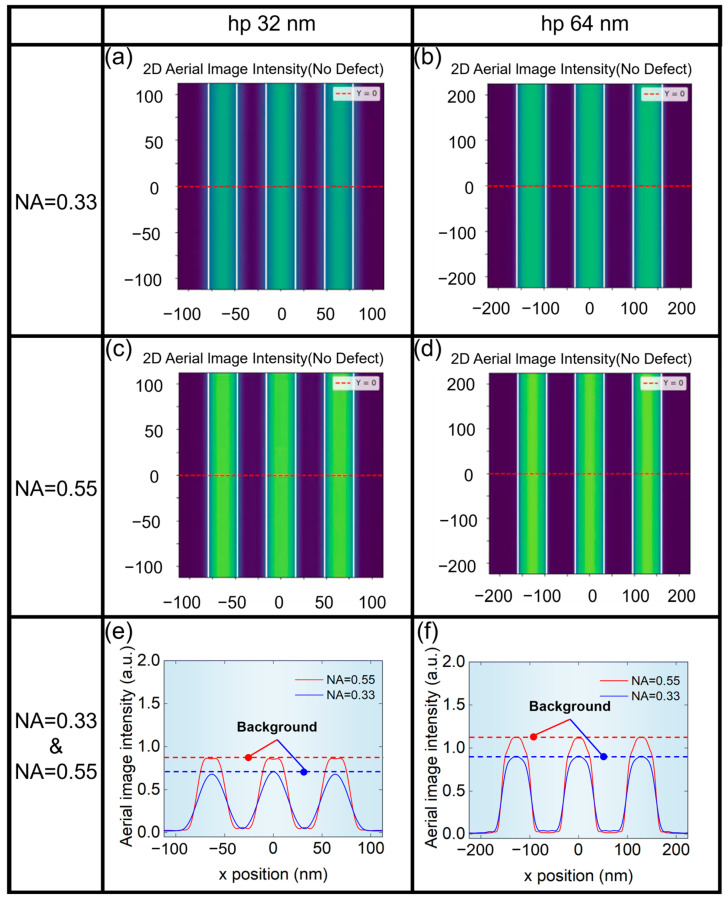
Aerial images of L/S pattern without phase defects. (**a**) hp 32 nm and (**b**) hp 64 nm at NA = 0.33. (**c**) hp 32 nm and (**d**) hp 64 nm at NA = 0.55. (**e**) hp 32 nm at NA = 0.33 and NA = 0.55. (**f**) hp 64 nm at NA = 0.33 and NA = 0.55.

**Figure 5 micromachines-16-01210-f005:**
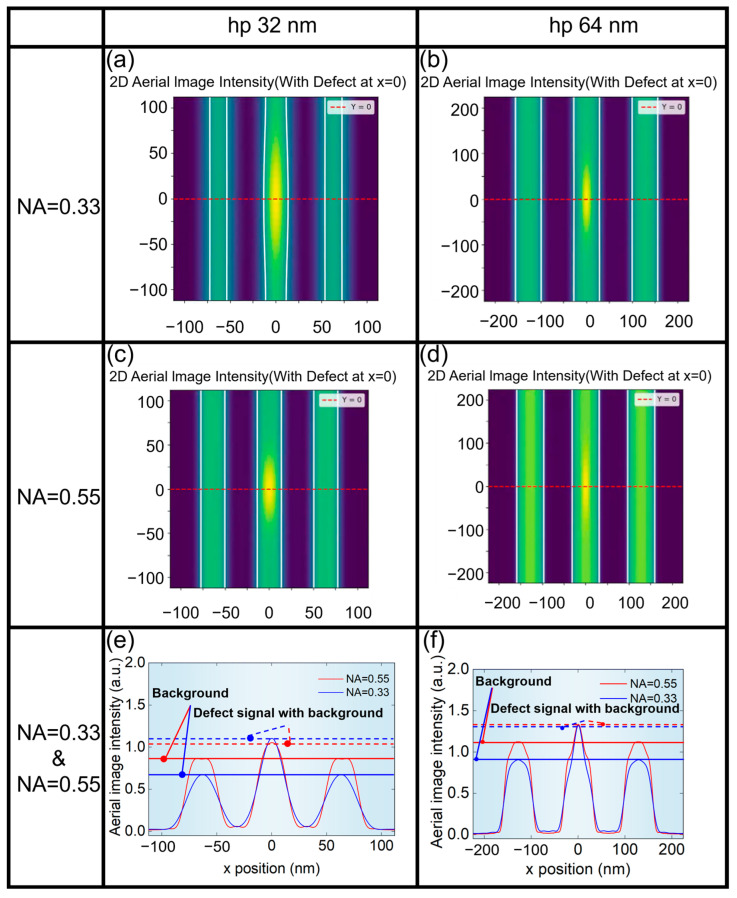
Aerial images of the L/S pattern affected by phase defects. (**a**) hp 32 nm and (**b**) hp 64 nm at NA = 0.33. (**c**) hp 32 nm and (**d**) hp 64 nm at NA = 0.55. (**e**) hp 32 nm at NA = 0.33 and NA = 0.55. (**f**) hp 64 nm at NA = 0.33 and NA = 0.55.

**Figure 6 micromachines-16-01210-f006:**
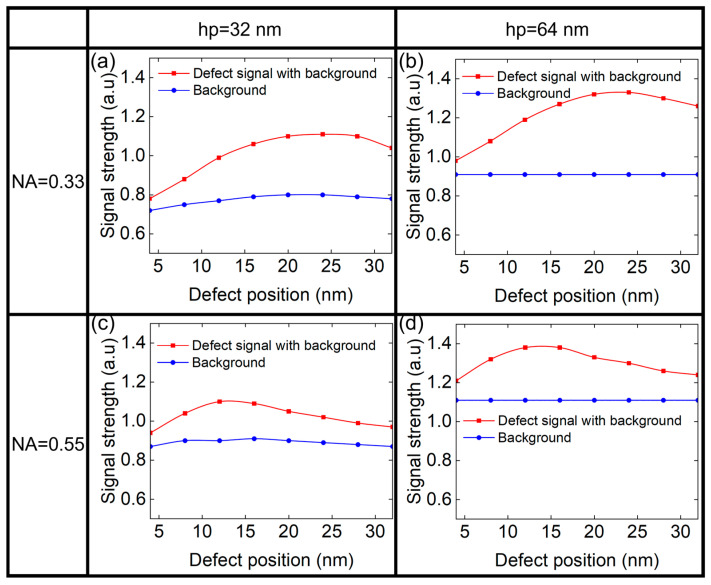
The defect signal and background signal of L/S pattern of (**a**) hp 32 nm and (**b**) hp 64 nm at NA = 0.33. The defect signal and background signal of L/S pattern of (**c**) hp 32 nm and (**d**) hp 64 nm at NA = 0.55.

**Figure 7 micromachines-16-01210-f007:**
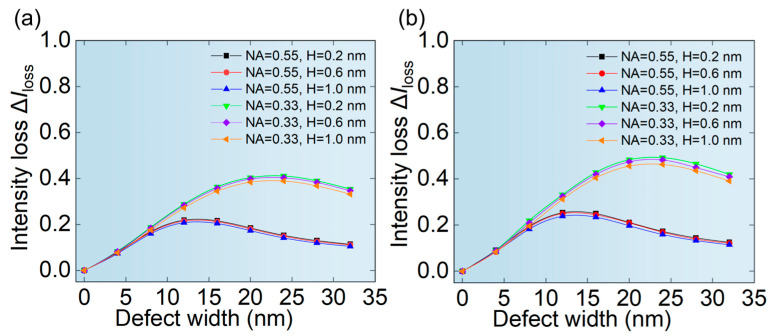
The relationship between the width of phase defects and the ∆*I*_loss_ of aerial images of (**a**) hp 32 nm L/S pattern and (**b**) hp 64 nm L/S pattern.

**Figure 8 micromachines-16-01210-f008:**
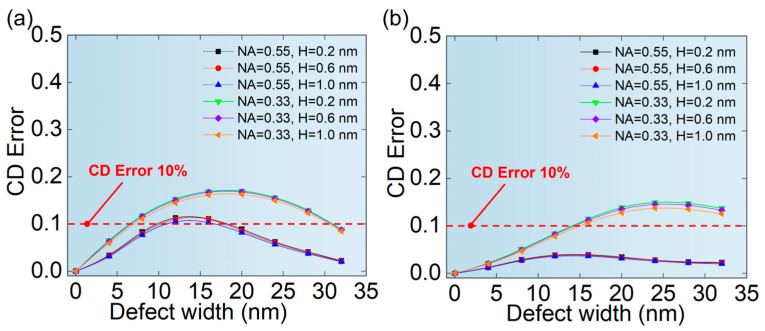
The relationship between the width of phase defects and the CD error of printed patterns of (**a**) hp 32 nm L/S pattern and (**b**) hp 64 nm L/S pattern.

**Figure 9 micromachines-16-01210-f009:**
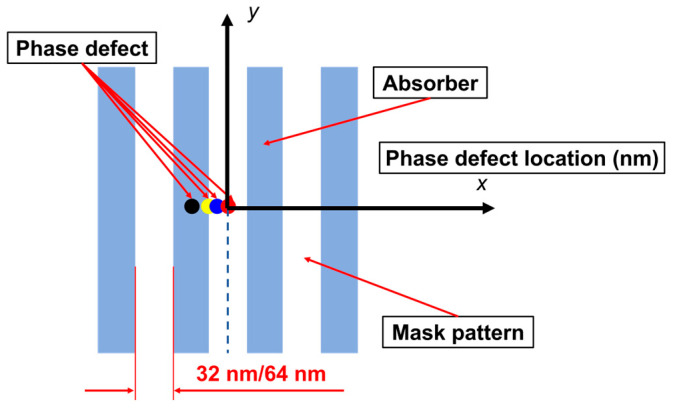
The schematic for the position of phase defects in the L/S pattern.

**Figure 10 micromachines-16-01210-f010:**
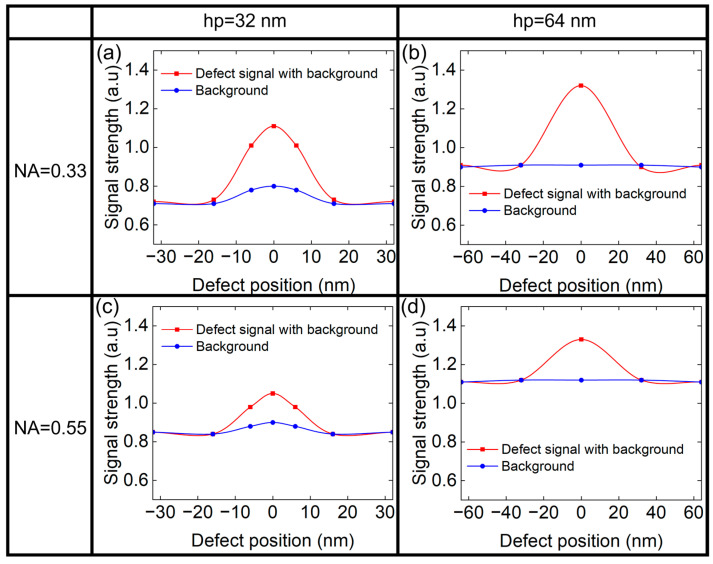
The defect signal and background signal of (**a**) hp 32 nm and (**b**) hp 64 nm at NA = 0.33 at different positions of the L/S pattern. The defect signal and background signal of (**c**) hp 32 nm and (**d**) hp 64 nm at NA = 0.55 at different positions of the L/S pattern.

**Figure 11 micromachines-16-01210-f011:**
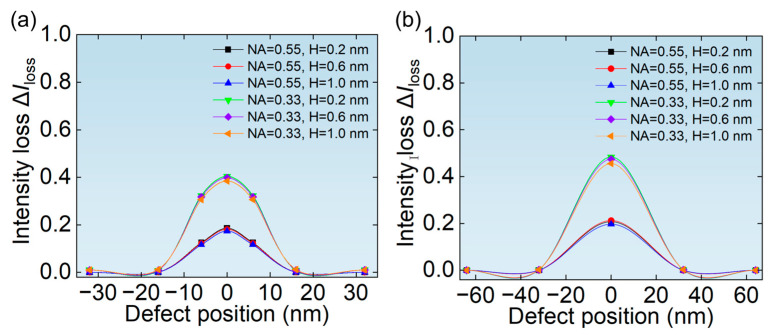
The relationship between the position of phase defects and the ∆*I*_loss_ of aerial images of (**a**) hp 32 nm L/S pattern and (**b**) hp 64 nm L/S pattern.

**Figure 12 micromachines-16-01210-f012:**
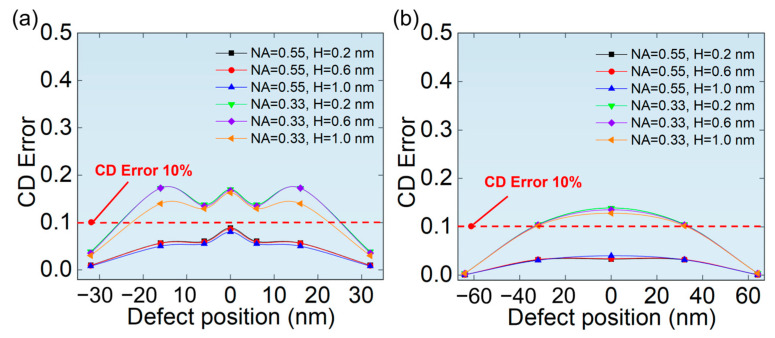
The relationship between the position of phase defects and the CD error of printed patterns of (**a**) hp 32 nm L/S pattern (**b**) hp 64 nm L/S pattern.

**Table 1 micromachines-16-01210-t001:** The optical properties of materials in EUV mask.

Name	Notation	Unit	Value
Thickness of the SiO_2_	$d_{{SiO}_{2}}$	nm	30
Refraction index of the SiO_2_ [27]	$n_{{SiO}_{2}}$	1	0.9782 + 0.0108*j*
Thickness of the Mo	*d* _Mo_	nm	2.78
Refraction index of the Mo [27]	*n* _Mo_	1	0.9238 + 0.0064*j*
Thickness of the Si	*d* _Si_	nm	4.17
Refraction index of the Si [27]	*n* _Si_	1	0.999 + 0.0018*j*
Thickness of the Ta	*d* _Ta_	nm	70
Refraction of the Ta [27]	*n* _Ta_	1	0.9429 + 0.0408*j*

**Table 2 micromachines-16-01210-t002:** The parameter of illumination system.

Name	Notation	Unit	Value
Illumination wavelength	λ	nm	13.5
Outer coherence factor [28]	*σ* _out_	1	0.8
Inner coherence factor [28]	*σ* _in_	1	0.6
Azimuth angle [33]	*φ*	°	0
Angle of rotate [33]	*θ*r	°	0
Angle of incidence [33]	*α*	°	6

## Data Availability

Data underlying the results presented in this paper are not publicly available at this time but may be obtained from the authors upon reasonable request.

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
