# Peer review of "Impact of Phase Defects on the Aerial Image in High NA Extreme Ultraviolet Lithography"

_micromachines, 2025, doi:10.3390/mi16111210_

Round 1

Reviewer 1 Report

Comments and Suggestions for Authors

Overall, the study is sound, and the conclusions are supported by the data. There are, however, some minor points that need attention. 

  1. The article title is not appropriate. First, the article did not discuss about "printability", which is included in the title. Second, the article elaborates on the impact of high NA and large hp on the imaging effects and CD errors of phase defects, and the title could be more impactful by better reflecting the novel aspect of this work.
  2. In Figure 1, the legend for the absorber should be Ta rather than TaN.
  3. In page 8, the figure 6 is not concise and some texts are not clear. What's the difference between line-dot and bar chart? If they contain the same information, just keep one type of the chart like the authors did in Figure 10. The legends for the red dots and cylindar is "background signal and defect signal" is also confusing.  I suggest using "defect signal" or "defect signal with background".
  4. In all figures, the text is not very clear. I suggest the authors using "Arial" font.

Author Response

Comments 1: The article title is not appropriate. First, the article did not discuss about “printability”, which is included in the title. Second, the article elaborates on the impact of high NA and large hp on the imaging effects and CD errors of phase defects, and the title could be more impactful by better reflecting the novel aspect of this work.

Response 1: Thank the reviewer for the valuable suggestion. In the revised manuscript, the term “printability” has been removed to avoid misrepresentation. We have updated the title to “Impact of phase defects on the aerial image in high NA extreme ultraviolet lithography”. This new title more accurately reflects the scope of the study by emphasizing the influence of high numerical aperture (high-NA) and varying half-pitch conditions on aerial image and critical dimension (CD) errors induced by phase defects. The modification has been made in the revised manuscript. 

Comments 2: In Figure 1, the legend for the absorber should be Ta rather than TaN.

Response 2: Thank the reviewer for the comment. In the revised manuscript, we have corrected the legend for the absorber in Figure 1 by replacing “TaN” with “Ta” to accurately reflect the material. The modification has been made in the revised manuscript.

Comments 3: In page 8, the figure 6 is not concise and some texts are not clear. What's the difference between line-dot and bar chart? If they contain the same information, just keep one type of the chart like the authors did in Figure 10. The legends for the red dots and cylindar is “background signal and defect signal” is also confusing. I suggest using “defect signal” or “defect signal with background”.

Response 3: Thank the reviewer for the comment. We agree that figure 6 was indeed not concise and that some texts were indeed unclear in the original manuscript. In the revised manuscript, we have updated the plotting format of figures 6 and 10 to be consistent, using a unified curve graphs to improve clarity and visual consistency. Figure 6 shows the background signal and the defect signal of different size depending on the NA and the hp of the printed patterns. In figure 6, the red curve represents the defect signal with background, while the blue curve indicates the background. The defect signal can be obtained as the difference between the two curves.

In the revised manuscript, figures 6 and 10 have been modified. We have also modified the corresponding discussion in the revised manuscript to make it clear to the readers, which is marked as red text.

Comments 4: In all figures, the text is not very clear. I suggest the authors using “Arial” font.

Response 4: Thank the reviewer for the valuable suggestion. In the revised manuscript, all figures have been updated to use “Arial” font, and font sizes have been slightly increased to improve text clarity and ensure typographic consistency throughout the paper. The modifications have been made in the revised manuscript.

Reviewer 2 Report

Comments and Suggestions for Authors

Manuscript ID: micromachines-3916853

Title: Analysis of Imaging Effects and Printability of Phase Defect in High-NA EUV Mask

Review Comments

This manuscript presents a numerical study investigating the impact of buried Gaussian phase defects on aerial image quality and critical dimension (CD) error in line/space patterns for High-NA (0.55) versus conventional NA (0.33) EUV lithography. Using rigorous coupled-wave analysis (RCWA) and the Hopkins model, the authors simulate effects across varying defect sizes (width) and positions. This work demonstrates a certain level of innovation, providing detailed content and substantial evidence.However, the manuscript still has the following issues that require specific attention:

  • This paper assumes a simplified Gaussian pit defect model where top and bottom heights and widths are equal. Real phase defects often have more complex, asymmetric shapes resulting from substrate pits/bumps and subsequent multilayer deposition. The study should explicitly discuss the limitations of this symmetric model and how it might influence the generalizability of the findings.
  • The introduction effectively reviews prior work but could better position the novelty of this study. Specifically, contrasting the findings more directly with the conflicting or complementary results from cited works would highlight the specific contribution of this paper in advancing the understanding of buried phase defects under High-NA conditions.
  • The observed fluctuation in CD error for hp 32 nm at NA=0.33 is attributed to the "shadow effect." While this is a known phenomenon in EUV due to oblique illumination, the explanation provided is somewhat qualitative. A more rigorous discussion, potentially supported by illustrations of light interaction with the absorber edge and the defect at different positions, would enhance understanding. Quantifying how the effective illumination intensity varies spatially due to the absorber's occlusion could provide deeper insight into the mechanism behind the double-peak CD error behavior.
  • The font size of the axis labels (both vertical and horizontal) in Figures 4e, 4f, 5e, 5f, 6, and 10 is too small, making them difficult to read. It is recommended that the authors adjust the font size appropriately for better clarity.

Therefore, I recommend that this manuscript be accepted for publication in the Micromachines after the authors address the minor revisions outlined above.

Author Response

Comments 1: This manuscript presents a numerical study investigating the impact of buried Gaussian phase defects on aerial image quality and critical dimension (CD) error in line/space patterns for High-NA (0.55) versus conventional NA (0.33) EUV lithography. Using rigorous coupled-wave analysis (RCWA) and the Hopkins model, the authors simulate effects across varying defect sizes (width) and positions. This work demonstrates a certain level of innovation, providing detailed content and substantial evidence. However, the manuscript still has the following issues that require specific attention:

This paper assumes a simplified Gaussian pit defect model where top and bottom heights and widths are equal. Real phase defects often have more complex, asymmetric shapes resulting from substrate pits/bumps and subsequent multilayer deposition. The study should explicitly discuss the limitations of this symmetric model and how it might influence the generalizability of the findings.

The introduction effectively reviews prior work but could better position the novelty of this study. Specifically, contrasting the findings more directly with the conflicting or complementary results from cited works would highlight the specific contribution of this paper in advancing the understanding of buried phase defects under High-NA conditions.

The observed fluctuation in CD error for hp 32 nm at NA=0.33 is attributed to the "shadow effect." While this is a known phenomenon in EUV due to oblique illumination, the explanation provided is somewhat qualitative. A more rigorous discussion, potentially supported by illustrations of light interaction with the absorber edge and the defect at different positions, would enhance understanding. Quantifying how the effective illumination intensity varies spatially due to the absorber's occlusion could provide deeper insight into the mechanism behind the double-peak CD error behavior.

The font size of the axis labels (both vertical and horizontal) in Figures 4e, 4f, 5e, 5f, 6, and 10 is too small, making them difficult to read. It is recommended that the authors adjust the font size appropriately for better clarity.

Response 1: Thank the reviewer for the valuable comment and suggestion. In the revised manuscript, all figures have been updated, and font sizes have been slightly increased to improve text clarity and ensure typographic consistency throughout the paper. The modifications have been made in the revised manuscript. 
